# Precision Medicine Highlights Dysregulation of the CDK4/6 Cell Cycle Regulatory Pathway in Pediatric, Adolescents and Young Adult Sarcomas

**DOI:** 10.3390/cancers14153611

**Published:** 2022-07-25

**Authors:** Farinaz Barghi, Harlan E. Shannon, M. Reza Saadatzadeh, Barbara J. Bailey, Niknam Riyahi, Khadijeh Bijangi-Vishehsaraei, Marissa Just, Michael J. Ferguson, Pankita H. Pandya, Karen E. Pollok

**Affiliations:** 1Department of Medical and Molecular Genetics, Indiana University School of Medicine, Indianapolis, IN 46202, USA; fbarghi@iu.edu; 2Department of Pediatrics, Herman B Wells Center for Pediatric Research, Indiana University School of Medicine, Indianapolis, IN 46202, USA; heshanno@iu.edu (H.E.S.); msaadatz@iu.edu (M.R.S.); bajbaile@iu.edu (B.J.B.); nriyahi@iu.edu (N.R.); khbijang@iu.edu (K.B.-V.); 3Department of Pediatrics, Hematology/Oncology, Indiana University School of Medicine, Indianapolis, IN 46202, USA; justm@iu.edu (M.J.); micjferg@iu.edu (M.J.F.); 4Department of Pharmacology and Toxicology, Indiana University School of Medicine, Indianapolis, IN 46202, USA

**Keywords:** oncology, pediatrics, precision medicine, sarcoma, osteosarcoma, rhabdomyosarcoma, Ewing sarcoma, CDK4, CDK6, cyclin D, therapeutic response biomarker

## Abstract

**Simple Summary:**

This review provides an overview of clinical features and current therapies in children, adolescents, and young adults (AYA) with sarcoma. It highlights the basic and clinical findings on the cyclin-dependent kinases 4 and 6 (CDK4/6) cell cycle regulatory pathway in the context of the precision medicine-based molecular profiles of the three most common types of pediatric and AYA sarcomas—osteosarcoma (OS), rhabdomyosarcoma (RMS), and Ewing sarcoma (EWS).

**Abstract:**

Despite improved therapeutic and clinical outcomes for patients with localized diseases, outcomes for pediatric and AYA sarcoma patients with high-grade or aggressive disease are still relatively poor. With advancements in next generation sequencing (NGS), precision medicine now provides a strategy to improve outcomes in patients with aggressive disease by identifying biomarkers of therapeutic sensitivity or resistance. The integration of NGS into clinical decision making not only increases the accuracy of diagnosis and prognosis, but also has the potential to identify effective and less toxic therapies for pediatric and AYA sarcomas. Genome and transcriptome profiling have detected dysregulation of the CDK4/6 cell cycle regulatory pathway in subpopulations of pediatric and AYA OS, RMS, and EWS. In these patients, the inhibition of CDK4/6 represents a promising precision medicine-guided therapy. There is a critical need, however, to identify novel and promising combination therapies to fight the development of resistance to CDK4/6 inhibition. In this review, we offer rationale and perspective on the promise and challenges of this therapeutic approach.

## 1. Introduction

Positive outcomes in the treatment of pediatric, as well as adolescent and young adult (AYA) cancers, have improved remarkably over the past several decades [1]. The first key to this success was the implementation of cooperative clinical research trial groups among academic centers that were focused on the treatment of pediatric and AYA cancers. This collaboration led to the generation of clinical data indicating patient response to treatment options from relatively large numbers of patients, which ultimately impacted therapeutic outcomes and increased long-term survival rates. Interdisciplinary multimodal therapy approaches such as polychemotherapy protocols and applying drug combinations sequentially was the second key to improving survival in pediatric patients [2]. This phenomenon is most evident in acute lymphoblastic leukemia (ALL), where survival in some subgroups of ALL is now up to 90% of patients [3]. Improvements in survival in solid tumors such as sarcoma have been slower, with the 5-year event-free survival (EFS) rate approaching 70%; however, 5-year survival for patients with metastatic disease still only remain between 20-30% [4]. This is compounded by the fact that sarcomas are quite rare and heterogeneous in children and AYA patients [5], making the testing of new therapies in adequately powered clinical trials extremely challenging [2]. Additionally, metastatic tumors harbor extremely complex genomic alterations—demonstrated by next-generation sequencing (NGS)—that can differ from the primary malignancy [6]. Moreover, these patients are at a very high risk of developing chronic health issues affecting long-term morbidity and mortality due to the intensity of front-line and salvage therapies [4]. Therefore, personalized treatment that focuses on specific actionable molecular alterations in sarcoma patients with high-risk signatures at diagnosis, or with metastatic and/or relapsed disease, have emerged as an opportunity to increase survival and improve quality of life [4]. While conventional chemotherapy can damage both cancer cells and normal tissues, treating sarcomas based on individual molecular profiles has the potential to specifically target the cancer cells and minimize the toxicity to normal tissues. Moreover, molecular signatures are increasingly used in diagnosis, as well as being used as biomarkers of therapeutic sensitivity and resistance. They also can be exploited to uncover additional intra-tumoral heterogeneity and, hence, redefine tumor subtype classifications [2]. By further refining classifications based on molecular signatures, tumor subtypes are being further subdivided into smaller categories, emphasizing the critical need to incorporate precision genomic approaches in guiding therapy selection [2].

Molecular analyses of pediatric and AYA sarcomas have led to the identification of cyclin-dependent kinases (CDK) CDK4 and CDK6 as a therapeutic targeting opportunity. Activation of the CDK4/6 pathway can be a powerful driver of sarcomagenesis [7]. It is a major cell cycle regulatory pathway in mammalian cells, regulating the progression of cells into the DNA synthesis (S phase) part of the cell cycle [7]. The objective of this review is two-fold. First, a brief overview of pediatric and AYA sarcoma clinical features and current therapies is provided. This is followed by an assessment of the basic and clinical findings that are already published in the literature on the CDK4/6 cell cycle regulatory pathway in the context of precision medicine-based molecular profiles in pediatric and AYA osteosarcoma (OS), Ewings Sarcoma (EWS), and rhabdomyosarcoma (RMS).

## 2. Pediatric and AYA Sarcomas

Sarcomas are rare among adult malignancies, representing only 8% of all the malignancies that are diagnosed in the United States each year [8,9,10]. Similarly, sarcoma is rare in children and adolescents, with OS, RMS, and EWS accounting for 1011 out of 16,850 diagnoses in this group in 2020 [10]. Though equally rare in childhood and adolescence, sarcomas are unfortunately the cause of 13% of cancer-related deaths in the US [4]. Additionally, it has been reported that approximately 1600 young adults are diagnosed with soft tissue sarcomas (STS) and bone sarcomas annually [11]. Sarcomas are of mesenchymal origin and can arise in bone or soft tissues. Among the bone sarcomas, OS and EWS are the most common types and represent 56% and 34% of bone sarcomas, respectively [1]. STS are further classified into rhabdomyosarcoma (RMS) and non-rhabdomyosarcoma [1]. Sarcomas are among the most “hard-to-cure malignancies” because of their aggressive biological behavior and occurrence at almost every anatomical site [9]. The epidemiology, clinical features, and cancer predisposition syndromes of the sarcomas have been reviewed elsewhere [11,12,13,14,15,16,17,18,19,20,21,22,23,24,25,26,27,28,29,30,31,32,33,34,35,36,37,38,39]. As mentioned above, this is also compounded by the rarity of these cancers, making it challenging to conduct clinical trials. Despite the administration of the optimized multimodal treatment, including the combination of surgery, chemotherapy, irradiation, immunotherapy, and/or targeted therapeutics, greater than 30% of sarcoma patients succumb to their disease [9]. For metastatic and relapsed sarcoma patients, the survival rates are abysmal, with less than 30% of patients alive several years after diagnosis [4]. Thus, the identification of actionable targets for these difficult-to-treat tumors is urgently needed. Technological advancements in NGS leading to precise molecular characterization of individual sarcomas has helped to identify and prioritize therapeutic targets and prognostic/predictive biomarkers for potential treatments [9]. The ultimate goal of precision medicine is to increase the validity of diagnosis and prognosis, as well as to identify the most effective and safe therapy for the treatment of pediatric and AYA sarcomas [2,9]. To achieve this goal, however, the molecular mechanisms that contribute to the development and pathogenesis of pediatric sarcomas require further investigation to identify targeted combination therapies and additional actionable therapeutic targets for future clinical use [4].

### 2.1. Osteosarcoma (OS)

#### Prognosis and the Standard of Care

OS is the most common primary bone cancer in children and AYA, with an overall incidence rate of 4.5 per million in patients under the age of 25 in the US [40]. Metastatic disease is found in up to 25% of OS patients, where the lungs are the most common site of disease spread [40]. The 5-year survival rates in localized and metastatic OS patients under the age of 25 are approximately 70% and 30%, respectively [13]. While metastasis is a major poor prognosis marker for OS, additional factors have been identified that contribute to poor prognosis in OS. These include large tumors (>10 cm), axial tumor locations, male gender, OS of the pelvis, and elevated alkaline phosphate levels at diagnosis [14,41]. The mutational landscape of OS is highly complex and differs remarkably between tumors. Unlike other pediatric sarcomas, OS lacks a specific chromosome translocation or recurrent driver mutations. Instead, it is characterized by a high level of chromosomal instabilities (CINs) such as copy number variations (CNVs) [42]. The genomic complexity of OS greatly complicates identifying therapeutic strategies for OS patients. Standard-of-care for OS patients is similar across most OS subtypes, which includes neoadjuvant chemotherapy, surgical resection of the primary tumor with wide margins, and adjuvant chemotherapy [12,14]. MAP (methotrexate, adriamycin, and cisplatin) chemotherapy is the most common initial treatment for OS [43]. Radiotherapy is not included in standard-of-care for OS patients, since these tumors are notoriously resistant to it [44]. The addition of chemotherapy regimens to surgical resection increased the 5-year overall survival from ~20% to up to 70% for patients with localized disease [14]. In the relapsed setting, there are no standardized second-line cytotoxic chemotherapies for OS. In terms of targeted agents, more investigations are needed to identify single and combinations of targeted agents for relapsed OS. The use of multi-receptor tyrosine kinase (RTK) inhibitors such as cabozantinib and regorafenib has recently exhibited some success in clinical trials [44,45].

### 2.2. Ewing Sarcoma (EWS)

#### Prognosis and the Standard of Care

EWS is the second most common primary bone malignancy in children and AYA patients and can be diagnosed as a primary soft tissue tumor with the incidence in children and adolescents of ~3 cases per 1,000,000 per year in the US. The anatomic site of tumor involvement, tumor size, and the effect of treatment on tumor necrosis are significant prognostic factors, but metastasis is the by far the most adverse prognostic factor in EWS patients [35]. In children and adolescents, patients with localized EWS have a 5-year overall survival of about 70%, but patients with metastases at diagnosis show a markedly lower overall survival (>30%) [33]. After both neoadjuvant and adjuvant chemotherapy, EWS 5-year overall survival may be lower in patients with pelvic tumors, large tumors (>8 cm), and incomplete tumor regression [46]. The main somatic driver mutation in EWS is the genetic translocation that fuses the EWSR1 gene on chromosome 22 with other genes, mainly the FLI1 gene, a member of the E26 transformation specific (ETS) gene family of transcription factors on chromosome 11 [47,48]. This fusion is found in 85–90% of EWS patients [49]. Therefore, the *EWSR1*/*FLI1* fusion gene functions as the major oncogene by acting as an aberrant transcription factor in Ewing sarcoma [48]. For example, E2F3, a member of the E2F family of transcription factors, is directly activated by the EWSR1/FLI1 fusion protein. The EWSR1/FLI1-mediated activation of E2F3 leads to activation of the genes that are involved in cell cycle progression [50]. In 10% of EWS cases, the *EWSR1* gene is also fused to the *ERG* gene, another member of the *ETS* gene family of transcription factors, on chromosome 22 which also results in an aberrant transcription factor [49]. Clinically, the suppression of fusion proteins has not been successfully achieved in EWS for more than 20 years; however, the promising inhibition of fusion proteins—especially EWS/FLI1—has been shown preclinically and is currently in clinical trials [51]. Standard-of-care for EWS patients involves multidisciplinary management with surgical resection; local radiation; chemotherapy with vincristine and doxorubicin; and cyclophosphamide with ifosfamide and etoposide [52].

### 2.3. Rhabdomyosarcoma (RMS)

#### Prognosis and the Standard of Care

RMS is the most common soft tissue sarcoma in children, with an overall incidence rate of approximately 4.5 patients per million in individuals younger than 20 years old in the US. Prognostic indicators in RMS, by which patient outcome is directly influenced, include patient age, histology subtype, tumor site, tumor size, and stage of the disease. It has been shown that patients younger than ten years who have localized tumors that measure <5 cm have a higher chance of survival [26,27,30]. Additionally, favorable prognosis is achieved when tumors occur at the following sites: head and neck, non-parameningeal orbital, paratesticular and genitourinary-non-bladder/prostate sites [27,30]. In RMS, 5-year overall survival is the worst for alveolar RMS (ARMS) patients, with approximately 25% and 30% 5-year survival, respectively. ERMS has the highest 5-year survival rate at approximately 70%. There are no significant differences in the survival between gender or race among all of the RMS subtypes [53].

The improvement in cure rates for localized RMS has increased from 25% to 70% and is largely due to multidisciplinary cooperative group trials and multimodal treatment protocols [30]. However, <30% of patients with the metastatic disease survive beyond 5 years, despite aggressive therapy [54]. RMS patients can also be classified genetically as fusion-positive or fusion-negative cases. Most ARMS cases (~80%) are fusion positive with chromosomal translocations of the *PAX* genes—*PAX3* on chromosome 2 or *PAX7* on chromosome 1, to *FOXO1* on chromosome 13 [55]. By resulting in aberrant transcription factors, these gene fusions lead to poor patient prognosis, as they are correlated with tumor aggressiveness [54,55]. Since developing small molecule inhibitors that are selective for fusion transcription factors is challenging, indirect inhibition is more promising by targeting the genes that are controlled by the fusion proteins [56]. For example, the amplification of *CDK4*, *MYCN*, and *MIR17HG* genes has been found predominantly in *PAX* gene fusion-positive (PFP) tumors, which can be considered as therapeutic targets in PFP RMS patients [54]. A new gene fusion involving a FET-family gene with *TFCP2* has also been recently identified in epithelioid- and spindle-cell RMS patients, revealing a new fusion-positive RMS subgroup of highly aggressive craniofacial intraosseous RMS [57,58]. In most intraosseous RMS cases, tumors usually destroy the bone cortex and invade the surrounding soft tissue. The most common bones affected are craniofacial bones [57]. Following a histological confirmation of RMS, almost all patients receive chemotherapy, including vincristine, actinomycin, and cyclophosphamide (VAC), before undergoing surgery and/or radiotherapy [59,60]. Neoadjuvant chemotherapy maximizes the chance of complete resection at surgery. However, complete resection is rarely possible, particularly in tumors with unfavorable sites, which highlights the importance of radiotherapy as standard of care in RMS [27]. Second-line treatment for relapsed RMS patients is not standardized but is based on institutional preference, which can include various combinations of cytotoxic drugs such as ifosfamide, doxorubicin, temsirolimus, and vincristine [60,61].

## 3. Precision Medicine and Targeted Therapies in Pediatric and AYA Sarcomas: Effects on Diagnosis and Prognosis

Overall, there is an unmet clinical need to develop more efficacious and well tolerated therapeutic strategies for all three common forms of sarcoma (OS, RMS, and EWS) in children and AYAs. The molecular profiling of tumor samples can be a valuable tool for identifying actionable targets and corresponding inhibitors that can be a component of the clinical treatment plan in pediatric and AYA sarcoma patients. The evolution of NGS approaches has revolutionized preclinical and clinical cancer research in recent years by providing a major step forward in advancing precision medicine in the clinic [62]. Actionable alterations that are potentially responsive to specific targeted therapies can be detected in patients with various types of tumors by using the most common NGS approaches, which include whole genome sequencing (WGS), whole exome sequencing (WES), cancer-specific gene panels, RNA-seq, and proteomics [63,64]. Identifying tumor-specific aberrations with the potential to offer rational personalized treatments has improved the precision medicine concept in the clinic [62]. This is exemplified by a growing interest in the establishment of multi-disciplinary precision medicine programs that are focused on pediatric cancers, including sarcomas [65,66,67]. However, precision-guided treatments are in their infancy, and much remains to be learned in this field regarding the translation of all knowledge to the clinic in pediatric and AYA sarcoma patients [4]. Recent studies have detected targetable alterations in sarcoma patients and have used the findings to guide treatment [9]. Dysregulation of the primary cell cycle regulators is one of the detected actionable alterations that are gained by multiple-precision genomics/transcriptomics programs in pediatric and AYA sarcoma patients, and they are discussed in detail in the next section of this article.

### 3.1. The CDK4/6 Cell Cycle Regulatory Pathway Dysregulation in Pediatric and AYA Sarcomas

Dysregulation of the Cyclin D–CDK4/6 axis through various mechanisms leads to uncontrolled cell proliferation, frequently shown in many types of cancer, which are thought to show greater dependency on CDK4/6 [68,69]. Cancer cells frequently bypass the RB-dependent restriction point in the G1 phase of the cell cycle, typically through alterations in cell cycle machinery genes that lead to constitutive Cyclin D–CDK4/6 activation, or through the loss of RB itself [70]. Loss of the p16^INK4A^ function is also a common event leading to dysregulation of the Cyclin D–CDK4/6 axis in cancer cells, as it can predict strong CDK4/6 dependence in cancer cells [68]. Increased growth factor-mediated mitogenic signaling through key oncogenic pathways, such as hyperactive PI3K and MAPK pathways, can also assist cancer cells in bypassing the RB-dependent restriction point in the G1 phase of the cell cycle by promoting Cyclin D-CDK4/6 activity (Figure 1) [69]. Cyclin D–CDK4/6 axis dysregulation has been reported in a subset of pediatric and AYA sarcoma patients. Precision medicine programs indicating dysregulation of this major cell cycle regulatory pathway in pediatric and AYA sarcoma patients are discussed below.

#### 3.1.1. Dysregulation of the CDK4/6 Cell Cycle Regulatory Pathway in Pediatric and AYA OS

Recent precision medicine trials (PMTs) using -omics approaches have demonstrated dysregulation of the CDK4/6 cell cycle regulatory pathway in OS patients (Table 1). The Individualized Therapy for Relapsed Malignancies in Childhood (INFORM) precision medicine study on children with high-risk relapsed/refractory malignancies, including OS, aimed to detect therapeutic targets individually. In this study, whole-exome, low-coverage whole-genome, and RNA sequencing, as well as methylation and expression microarray analyses, were used. The results showed that in two out of four OS patients ≤ 20 years of age, CDK4/6 pathway-associated therapeutic targets, including either G91A point mutation in *CDKN2B* (predicted but not proven to affect CDKN2B function) or *CDK4* amplification, were detected. CDK4 inhibition was the matched targeted therapy for these genomic alterations in these patients [71]. The Precision in Pediatric Sequencing Program is a PMT in which the prospective clinical WES of patient-matched tumor-normal samples and the RNA-seq of tumors were conducted to identify potential targetable variants in patients who were initially diagnosed with pediatric malignancies, including OS, with the age range of 2 weeks to 26 years. In one out of the six OS patients who were enrolled in this study, *CCNE1* over-expression was detected as a potential therapeutic biomarker for the use of a CDK4/6 inhibitor. *RB* splice mutation was also detected as a clinically impactful germline alteration in one out of the six OS patients in this study [72]. In the individualized cancer therapy (iCat) trial, the feasibility of identifying actionable alterations was assessed in patients aged ≤30 years with extracranial solid tumors, including OS patients. In this trial, mutations were detected by Sequenom assay or targeted NGS, and copy number was assessed by comparative genomic hybridization (aCGH). In addition, RNA seq was performed in the trial if sufficient tumor specimen was available. *CCNE1* high copy number gain was identified as a potentially actionable in two out of the 11 OS patients who were enrolled in the trial [73]. Dinaciclib, which selectively inhibits cyclin-dependent kinases CDK1, CDK2, CDK5, and CDK9 [74], was recommended as a targeted agent to inhibit the CCNE1-mediated activation of CDK2 [73]. *CCND1* high copy number gain was also detected in one of the OS patients. For this actionable alteration, CDK4/6 inhibitor ribociclib was recommended as the targeted therapy [73]. In the molecular biology tumor board (MBB) program at Institute Curie, tumor profiling was conducted using panel-based NGS and array-comparative genomic hybridization on patients with a poor prognosis or relapsed/refractory solid tumors, including OS. In this study, in one out of four OS patients under the age of 19, *CDKN2A* homozygous deletion was identified as potentially actionable alteration, for whom a CDK4/6 inhibitor was recommended [75]. In the TRICEPS study (the personalized targeted therapy in refractory or relapsed cancer in childhood study), molecular profiling of pediatric and adolescent patients at the age of ≤18 was achieved by the WES of a matched tumor to normal germline samples and RNA sequencing of the tumor. In two out of the seven OS who were patients enrolled in this study, *CDKN2A* exhibited alterations, including either *CDKN2A* copy loss or *CDKN2A* homozygous deletion; one OS patient had *RB* copy loss [76]. In a phase 1 clinical trial program at MD Anderson Cancer Center, genomic profiling results gained by NGS were analyzed on advanced sarcoma patients, including OS patients with the age range of 8–76 years. In one out of 11 OS patients, CDK4 amplification was one of the actionable alterations found [77]. Based on an extensive panel of NGS data that were obtained from 67 pediatric and adult OS patients enrolled in a PMT by Memorial Sloan-Kettering Cancer Center (MSKCC), *CDK4* amplification in 13.4% of cases, and *CDKN2A/B* deletion/mutation in 26.9% of cases were identified as potentially actionable alterations for CDK4/6 inhibitors. No significant differences were identified between pediatric and adult OS patients in the frequency of the potentially actionable alterations [78]. In the Zero Childhood Cancer Program, 252 tumors from high-risk pediatric and AYA patients, including OS patients, were molecularly profiled using WGS and RNA-seq. Among the eight OS patients who were enrolled in this PMT, overexpression and CNV of *CCNE1* were identified in one case. Moreover, in two OS patients, somatic CNV and SV were detected in *RB* [67]. CDK4/6 inhibition was among the recommended therapies in this study for these patients [67]. The St. Jude Children’s Research Hospital-Washington University Pediatric Cancer Genome Project characterized the genomic landscape of OS by performing WGS on 34 pediatric and AYA OS tumors and matched normal tissue samples. The data showed that *RB* underwent significant recurrent somatic alterations in 10 out of 34 cases. *RB* was mutated by both point mutations and structural variations (SVs) in these cases. In addition, in one patient, the SV of *CCND3* was detected [79]. In another study, “CRB cancer des Hôpitaux de Toulouse; BB-0033-00014”, WGS was performed on tumor samples from seven high-grade OS patients who were aged ≤20 at diagnosis, who were matched with non-tumor tissues from the same patients. A stop-gain mutation in the *RB* gene was identified in one case. In this patient, a low number of secondary mutations in other genes were observed, compared to patients with the wild-type *RB* gene [80]. As part of another study, 59 tumor/normal pairs of pediatric OS samples were examined using WES, WGS, and RNA-seq. *CDKN2A/B* deletions were identified as potentially actionable alterations in 2 of 20 OS patients. These two patients’ tumor samples were *RB*-proficient, providing rationale for treatment with a CDK4/6 inhibitor [81]. In the Molecular Screening for Cancer Treatment Optimization (MOSCATO-01) trial on pediatric and AYA patients with recurrent or refractory solid tumors aged under 23 at diagnosis, four OS patients were among the patients who were enrolled in the trial. The tumor samples were analyzed by CGH array, NGS, WES, and RNA seq. *RB* deletion was detected in one OS patient [82].

#### 3.1.2. Dysregulation of the CDK4/6 Cell Cycle Regulatory Pathway in Pediatric and AYA RMS

Recent PMTs have demonstrated dysregulation of the CDK4/6 cell cycle regulatory pathway in a population of RMS patients (Table 1). In the Zero Childhood Cancer Program, a PMT with 17 RMS patients (11 fusion-positive (FP) and six fusion-negative (FN) patients) out of 252 pediatric and AYA high-risk cancer patients; dysregulation of Cyclin D–CDK4/6 axis associated genes, including overexpression of *CDK4*, *CDKN2A/B, CCND3* or *CCNE1*; and CNVs of *CDK4, CDKN2A/B, CCND2,* or *CCND3* were identified as therapeutic response biomarkers for CDK4/6 inhibition in ~45% of FP patients and in ~30% of FN RMS patients. CDK4/6 inhibition was among the recommended therapies for cell-cycle associated dysregulation [67]. In the INFORM study, 33% (two out of six) of cases had either *CDKN2A/B* deletion or a *CDK4* copy number gain in RMS patients who were 15 years or younger and diagnosed with either ERMS or ARMS. CDK4 inhibition as a matching targeted therapy was suggested for these patients [71]. For the iCat trial with patients aged 30 or younger who were diagnosed with extracranial solid tumors, including RMS tumors, *CDKN2A/B* deletion was observed as a potential biomarker of therapeutic response to CDK4/6 inhibitor therapy in one out of 11 ERMS patients. The CDK4/6 inhibitor, ribociclib, was recommended as targeted therapy for this patient [73]. In the MOSCATO-01 trial with 12 RMS patients, *CDKN2A* deletion was detected in three ERMS patients. CDK inhibition was recommended as a precision genomic guided therapy for the patients with this alteration [82]. ClinOmics Program, a clinical genomics study of children and AYAs with relapsed and refractory cancers aged 25 or younger, examined the feasibility of genome-guided precision therapy in the patients, including RMS, by performing a combination of WES, RNA-seq, and high-density single-nucleotide polymorphism array analysis of the tumor. In one out of three RMS patients who were enrolled in the study, somatic alterations of *CDKN2A* and *RB* were detected, where *CDKN2A* and *RB* underwent the loss of heterozygosity, leading to cell cycle dysregulation [83]. In the MBB trial, among four RMS patients aged under 19, *CDK4* amplification was detected as a potentially actionable alteration in one patient who was diagnosed with metastatic ARMS. This patient received the CDK4/6 inhibitor palbociclib and showed partial remission. However, the therapy was stopped after two months because of disease progression, emphasizing the need for a combination therapy approach and raising the question of whether earlier intervention may be the optimal strategy in the future [75]. In the TRICEPS study, *CDKN2A* point mutation was detected in one out of four RMS patients at 18 years or younger who were enrolled in the study. *RB* point mutation and loss of heterozygosity were identified in another patient who was diagnosed with ARMS [76]. In a phase 1 clinical trial program at MD Anderson Cancer Center, *CDK4* amplification was identified as an actionable gene alteration, defined as a gene alteration that could be targeted by CDK4/6 inhibitor palbociclib in 2 of 8 RMS cases. All the patients who were enrolled in this trial were diagnosed with advanced sarcoma (ages 8–76 years) [77]. A comparative genomic hybridization array analysis of the primary tumor from a 27-year-old patient with relapsed and chemotherapy-refractory ARMS identified the highest level of amplification in the 12q13–14 region—more specifically, *CDK4* was highly amplified in this region. In this study, *CDK4* inhibition was suggested as a potential therapeutic strategy for RMS [84]. In another study, array-CGH demonstrated copy number alteration of *CDKN2A* as a recurrent mutation in RMS patients, including pediatric and adult patients with FET-TFCP2 fusion. *CDKN2A* homozygous deletion was identified in 90% of these cases [57].

#### 3.1.3. Dysregulation of the CDK4/CDK6 Cell Cycle Regulatory Pathway in Pediatric and AYA EWS

Recent PMTs have also demonstrated dysregulation of the CDK4/6 cell cycle regulatory pathway in EWS patients (Table 1). In the INFORM precision medicine study, in 45.5% of (5/11) EWS patients aged 28 or younger, one of the following genes associated with Cyclin D–CDK4/6 axis was dysregulated, including the amplification of *CCND2*, overexpression of *CCND1*, or *CDKN2A/B* deletion. In one patient, both the overexpression of *CCND1* and *CDKN2A/B* deletion were identified. All these patients were EWS/ETS (either *EWSR1/FLI1* or *EWSR1/ERG*) fusion positive. CDK inhibitors such as CDK4/6 inhibitors were suggested for these patients as precision genomics (PG)- directed therapy [71]. In the iCat trial with patients aged ≤30 who were diagnosed with extracranial solid tumors, including EWS tumors, either two copy losses of *CDKN2A/B* or a single copy number gain of *CCND1* was identified in two out of 12 EWS patients. The iCat recommendation was CDK4/6 inhibitor ribociclib for these patients [73]. EWS patients were included in the ClinOmics Program for children and AYAs with relapsed and refractory cancers aged ≤25. In approximately 10% of EWS patients who were positive for *EWSR1/FLI1* fusion, *CDKN2A* homozygous loss was identified as a potential biomarker of therapeutic response to CDK4/6 inhibitor therapy [83]. In the TRICEPS study, in one of two EWS patients aged ≤18, *CDKN2A* copy loss was identified as a potentially actionable somatic alteration. This patient was positive for *EWSR1/FLI1* fusion [76]. In one of three *EWSR1* fusion-positive EWS patients enrolled in the phase 1 clinical trial program at MD Anderson Cancer Center, *CDK4* amplification was detected. *CDK4* amplification was one of the most actionable alterations in this study, which was defined as a gene alteration that could be directly targeted by CDK4/6 inhibitor palbociclib. All patients with different advanced sarcomas who were enrolled in this study were 8–76 years old [77]. The Zero Childhood Cancer Program reported that data on 18 EWS patients (out of 252 pediatric and AYA high-risk cancer patients who were enrolled in the study) demonstrated the homozygous deletion of *CDKN2A/B* as a potential biomarker of therapeutic response to CDK4/6 inhibition in three of the EWS patients. One of these three patients was FP for *EWSR1/FLI1*. In this study, CDK4/6 inhibition was recommended as a targeted therapy for these patients with deletion of *CDKN2A/B* [67]. *CDKN2A/B* deletions were demonstrated in four out of five EWS patients aged under 23 who underwent copy-number analysis and were enrolled in the MOSCATO-01 trial. Two of these patients with *CDKN2A/B* deletions were *EWS/FLI1* FP EWS patients. CDK4 inhibition therapy was recommended as targeted therapy for these patients [82]. In another PMT on pediatric and AYA EWS patients aged between 4 months and 21 years old, the molecular profiles of the tumors from the patients were obtained using WES, WGS, SNP array, and RNA-seq. *EWS/ETS* rearrangements—mostly *EWSR1/FLI1*—were identified in these patients. Focal deletions at chromosome 9p21.3, which was consistent with a loss of *CDKN2A*, occurred in 22% of the EWS tumors using a combination of SNP array and WES [85].

Altogether, these studies indicate that -omics analyses are critical in providing guidance on whether a CDK4/6 inhibitor should be considered as a potential targeted therapy in these three common types of pediatric and AYA sarcomas.

## 4. Inhibition of Cyclin-Dependent Kinases 4/6 (CDK4/6) as a Source of Therapeutic Targets in Pediatric and AYA Sarcomas Based on Precision Genomics/Transcriptomics Profiling

### 4.1. Cyclin D–CDK4/6 Axis Role in Cell Cycle Regulation

Dysregulation of the cell cycle is one of cancer’s hallmarks and provides numerous options for therapeutic intervention [68]. CDK4 and CDK6 are major cell cycle regulators in mammalian cells, where they induce the progression of cells into the DNA synthetic (S) phase of the cell cycle [70]. In the first gap phase (G1) of the cell cycle, the enzymatic activity of CDK4 and CDK6 are regulated positively by D-type cyclins (D1, D2, and D3), expressed in response to multiple extracellular signals such as stimulatory mitogens, cell–cell interaction, and differentiation inducers [86,87]. The three D-type cyclins are expressed, alone or together, in different cell lineages, which may be due to tissue-specific aspects of normal physiology underlying the differential expression of the D-type cyclins [86]. The cyclins form a holoenzyme complex, assembled with CDK4 or CDK6 (Cyclin D-CDK4/6 complex) [87]. The Cyclin D-CDK4/6 complex is not yet enzymatically fully activated, unless by CDK-activating kinase (CAK)-mediated activating phosphorylation of CDK4/6 [88]. CDK4 and CDK6 have mostly overlapping functions. Both can associate with all three D-type cyclins, as they share 71% amino acid identity [89]. However, it has been shown that CDK4 and CDK6 can differentially regulate cancer progression through various mechanisms in breast, prostate, and pancreatic cancer models, with the CDK4-dependent regulation of pro-metastatic inflammatory pathways, and the CDK6-dependent control of the genes that are involved in DNA replication and repair mechanisms. In these models, silencing CDK6 but not CDK4 led to the induction of DNA damage and defective DNA repair. In contrast, silencing CDK4 but not CDK6 enriched the TNF signaling pathway that is involved in the regulation of proinflammatory cytokines and chemokines [90]. It is currently not known if CDK4 and CDK6 differential regulation is operative in sarcomas, and it is an area that is worthy of investigation.

The Cyclin D-CDK4/6 complex regulates interactions between the RB protein and E2F transcription factors [69]. Among the seven E2F family members (E2F1-5, 2A7E, and E2F7), three of them, E2F1-3, are transcriptional activators [91]. Upon the direct interaction of E2F1-3 with RB, the transcriptional activity of E2Fs is inhibited, leading to a pause in cell cycle transition [91]. The expression level of E2F1-3 differs at various cell cycle phases; E2F3 expression is increased in early-G1 to mid-G1, while the increased expression of E2F1 and E2F2 occurs at the G1/S boundary [91]. E2F transcriptional activity is not only inhibited by the direct binding to RB, but also by the RB-mediated recruitment of histone deacetylase (HDAC) to the RB-E2F complex [92]. Initially, upon Cyclin D-CDK4/6-mediated RB phosphorylation, RB affinity to E2F and HDAC is reduced, promoting the transcription of genes such as Cyclin E2 [92]. Subsequently, the Cyclin E2-CDK2 complex hyperphosphorylates RB which induces full dissociation of the RB-E2F complex and the transcription of genes encoding proteins playing critical roles in the next phases of the cell cycle, such as Cyclin A2, Cyclin B1 and others involved in DNA replication and repair [92,93,94]. Cyclin E2-CDK2 exemplifies that the cell cycle is not only positively regulated by Cyclin D-CDK4/6, but also by other Cyclin-CDKs (Figure 1).

CDK4/6 can also regulate the cell cycle independently of RB. CDK4/6 initiates the phosphorylation of FOXM1 in G1, leading to the transcriptional activation of FOXM1, which is required for the expression of the genes that are involved in cell cycle progression [95]. CDK4 negatively regulates SMAD2/3, components of the TGFβ signaling cascade that exhibit anti-proliferative activity [96]. Moreover, the CDK4-mediated phosphorylation of SMAD3 leads to its inhibition and the release of G1 arrest. Therefore, diminishing SMAD3 activity by CDK4-mediated phosphorylation could contribute to tumorigenesis [97].

CDK/cyclin activity is negatively regulated by CDK inhibitors (CKIs) and is classified into two families of cell cycle inhibitors, including the CDK-interacting protein/kinase inhibitory protein (CIP/KIP) family comprising p21^CIP1^, p27^KIP1^ and p57^KIP2^, and the inhibitor of cyclin-dependent kinase 4 (INK4) family, including p16^INK4A^, p15^INK4B^, p18^INK4C^, and p19^INK4D^ [98,99]. The INK4 family of proteins specifically inhibits Cyclin D-CDK4/6 activity by directly binding to CDK4/6 to inhibit the formation of CDK4/6-cyclin D1, D2, and D3 complexes [98]. In contrast to the INK family, the CIP/KIP proteins can bind to all CDKs driving the cell cycle, including CDK1, CDK2, and CDK4/6 [98]. For example, p21^CIP1^ (inhibitor of CDK1 and CDK2) can bind to Cyclin D-CDK4/6 complex and relieve CDK2, further increasing its binding to Cyclin E2 and cell cycle progression [100]. p16^INK4A^ and p15^INK4B^, encoded by *CDKN2A* and *CDKN2B*, respectively, play significant roles as tumor suppressors in the INK4 family [101].

Dysregulation of the Cyclin D–CDK4/6 axis through various mechanisms, such as the gene amplification of positive regulators, or gene rearrangement; loss of negative regulators; epigenetic alterations; and point mutations in significant Cyclin D–CDK4/6 pathway components, leads to uncontrolled cell proliferation frequently shown in many types of cancer including sarcomas [69,102]. Therefore, because of the significant role of CDK4/6 activity in cancer cells, CDK4/6 inhibitors have emerged as well validated targeted therapy for cancer treatment [69]. As discussed later in this review, alterations leading to Cyclin D-CDK4/6 axis dysregulation could confer resistance or sensitivity to CDK4/6 inhibitors, an important point for the stratification of patients for CDK4/6 inhibitor therapy.

### 4.2. Selective CDK4/6 Inhibitors: Palbociclib, Abemaciclib, Ribociclib, Trilaciclib and Dalpiciclib

In addition to CDK4 and CDK6, other CDKs also play potential roles in cell cycle regulation. To date, 21 different CDKs have been identified in the human body that can interact with 29 different cyclins [103,104]. Therefore, the administration of pan-CDK inhibitors could cause significant toxicity, since multiple CDKs that are essential for the function of normal cells could also be inhibited [103,104]. The first-generation pan-CDK inhibitors such as flavopiridol and roscovitine lacked selectivity among the CDK family members [105]. The second generation including, dinaciclib, P276-00, AT7519, TG02, roniciclib, and RGB-286638 were developed based on the first-generation group but with increasing selectivity against CDK1 and CDK2 to reduce the off-target risks of CDK inhibitors [105,106]. These two generations of CDK inhibitors demonstrated limited efficacy and tolerable toxicity in clinical trials [86]. A third generation of CDK inhibitors with high selectivity for CDK4/6 was subsequently developed with the goal of reducing the risk of toxicity, while improving efficacy [107]. The high selectivity of CDK inhibitors to CDK4/6 is associated with reduced toxicity in normal cells. In addition, Santamaria et al. demonstrated that the genetic knockout of CDK4/6 in mice embryos did not interfere with normal mouse embryonic fibroblast cells and organogenesis, suggesting that the selective targeting of CDK4/6 should be tolerated. In these embryos, the compensatory effect of CDK1 was sufficient for the phosphorylation of RB and the expression of the genes that were regulated by E2F transcription factors [108]. To date, there are four U.S. Food and Drug Administration (FDA)-approved third-generation CDK4/6 inhibitors: palbociclib (Ibrance, PD0332991); ribociclib (Kisqali, LEE011); abemaciclib (Verzenio, LY2835219); and trilaciclib (Cosela, G1T28), all of which are employed in the clinic and have demonstrated some degree of clinical efficacy [106,109]. Palbociclib was the first orally bioavailable, highly selective CDK4/6 inhibitor and approved by the U.S. FDA in 2015 [110]. Subsequently, in 2017, ribociclib and abemaciclib were approved by the U.S. FDA [74,75]. Trilaciclib is the most recent CDK4/6 inhibitor to be approved by the U.S. FDA in 2021 [109]. As anticipated, the four selective CDK4/6 inhibitors suppress RB phosphorylation, thus causing G1 arrest [106,109]. There is also emerging evidence that CDK4/6 inhibition can block metabolic pathways, potentially leading to the control of tumor progression that is independent of RB status [111,112]. In addition, CDK4/6 inhibition is also very attractive, as it can influence the tumor microenvironment by impacting cell-mediated immunity and enhancing anti-tumor responses. For example, CDK4/6 inhibition increased IL-2 and CD4+ T-cell activation [113] and blocked the growth of immunosuppressive regulatory T cells [114]. A cellular assessment of the CDK4/6 inhibitors that measured biochemical potency and drug affinity demonstrated apparent differences among palbociclib, abemaciclib, and ribociclib. Palbociclib showed equivalent potency toward CDK4/Cyclin D3 and CDK6/Cyclin D1; however, both abemaciclib and ribociclib had significantly more CDK4/Cyclin D3 potency [115]. The biochemical profiling of trilaciclib demonstrated similar potency toward CDK4/Cyclin D1 and CDK6/Cyclin D3 [116]. FDA-approved palbociclib, abemaciclib, and ribociclib are currently used for the treatment of hormone receptor (HR)-positive, human epidermal growth factor receptor 2 (HER2)-negative (HR+/HER2-) advanced or metastatic breast cancer, in combination with endocrine therapy [107,117,118]. Among the three CDK4/6 inhibitors, abemaciclib is the only one that shows promising efficacy as a single agent to treat HR+ breast cancer [119]. In addition to clinical trials for breast cancer, the three CDK4/6 inhibitors underwent and are still undergoing testing in several clinical trials for other types of cancer, both as a single drug and in combination therapy [107]. Specifically, for pediatric sarcomas, there are several ongoing clinical trials to investigate the efficacy and safety of these CDK4/6 inhibitors (Table 2). Despite the similar mechanism of action for the first three CDK4/6 inhibitors, palbociclib, abemaciclib, and ribociclib, they exhibit different toxicity profiles and dose-delivery schedules. Since the half-lives of palbociclib and ribociclib are more than 24 h, they are dosed daily. Due to their myelosuppressive effects, daily dosing is non-continuous. Administration is for 21 days followed by a break of one week for neutrophil count recovery. In comparison, the half-life of abemaciclib is <24 h. It induces relatively less myelosuppression and is dosed twice daily [120] Palbociclib and ribociclib have minimal GI toxicities compared to abemaciclib, which has extensive GI toxicities [121]. However, abemaciclib-mediated bone marrow toxicity is less serious than palbociclib and ribociclib-mediated bone marrow toxicities [120]. The rate of hematological toxicity, mainly neutropenia, is higher for palbociclib and ribociclib [122]. Moreover, it has been shown that ribociclib can induce hepatic toxicity [123]. The most recently U.S. FDA-approved CDK4/6 inhibitor, trilaciclib, has been used to reduce the rate of chemotherapy-induced myelosuppression in patients with small-cell lung cancer. The administration of trilaciclib prior to chemotherapy could transiently induce G1 arrest of the cell cycle to prevent chemotherapy-induced myelosuppression [109]. Trilaciclib is well tolerated in the patients. Its administration before chemotherapy is not associated with a clinically relevant toxicity increase, and it significantly reduces chemotherapy-related toxicity [124]. To date, there are no clinical trials to investigate the efficacy and safety of trilaciclib in pediatric and AYA sarcomas. In addition to four U.S. FDA-approved CDK4/6 inhibitors, a novel CDK4/6 inhibitor, dalpiciclib (SHR6390), is approved by China’s National Medical Products Administration (NMPA) for the treatment of HR+ and HER2- recurrent or metastatic breast cancer with endocrine agents [125,126]. Dalpiciclib has a similar potency against CDK4 and CDK6 [126]. The oral administration of dalpiciclib every three weeks with a one-week break in patients with advanced breast cancer was well-tolerated [126].

#### 4.2.1. Intrinsic and Acquired Mechanisms of Resistance to CDK4/6 Inhibitors

While the selective CDK4/6 inhibitors show promising targeted therapy for cancer treatment, the onset of therapeutic resistance to these agents through both intrinsic and acquired molecular mechanisms has been encountered in cancer treatment. The development of resistance to CDK4/6 inhibitors leads to their suboptimal cytostatic effects, highlighting a need for combination therapy to evoke the cell death response [127,128].

##### Markers of Intrinsic Resistance and Sensitivity to CDK4/6 Inhibitors

Intrinsic mechanisms of resistance and sensitivity to CDK4/6 inhibitors are associated with alterations in specific cell cycle genes, which can serve as biomarkers of therapeutic resistance or sensitivity. Lack of RB, the primary target of CDK4/6, is a crucial contributor to CDK4/6 inhibitors’ intrinsic resistance [129]. Tumor cells with loss, or a low expression of *RB* are generally more resistant to CDK4/6 inhibitors than RB-proficient cells [68]. As a result, the vast majority of studies on CDK4/6 inhibitors have focused on RB-proficient cancer cells. In addition, alterations in other genes such as *CDK4*, *CCND* (the gene that encodes Cyclin D), *CDKN2A*, *E2F*, and *CDK2* can also contribute to CDK4/6 inhibitors’ intrinsic resistance [129]. The mentioned gene alterations are discussed in the specific studies below.

Konecny et al. studied the effect of palbociclib on 40 ovarian cancer cell lines and demonstrated that the copy number loss of *CDKN2A/B* and *RB* correlates with sensitivity and resistance to palbociclib, respectively [130]. A transcript microarray analysis of both palbociclib-sensitive and palbociclib-resistant human breast cancer cell lines identified 450 genes that were differentially expressed between sensitive cell lines (palbociclib IC50 < 150 nM) and resistant cell lines (palbociclib IC50 > 1000 nM). An increased expression of RB and *CCND1* and a decreased expression of CDKN2A were identified in association with sensitivity to the effect of palbociclib [131]. A phase II trial of palbociclib in liposarcoma patients with *CDK4* amplification and *RB* expression revealed that the sensitivity of these patients to palbociclib was associated with a favorable progression-free survival rate [132]. However, in FP RMS cell lines, resistance to ribociclib was evident in *CDK4*-amplified versus *CDK4*-wildtype cell lines. This differential responsiveness to ribociclib was also demonstrated in xenograft models of *CDK4*-amplified and wildtype FP RMS [133], implying that high levels of CDK4 may not be optimally inhibited. In addition, in FP RMS cell lines with a higher level of RB expression, increased sensitivity to ribociclib was shown, compared to FN RMS cell lines with a lower level of RB expression [133]. Consistent with the higher level of RB expression in FP RMS cell lines [133], *CDK4* amplification has also been demonstrated predominantly in FP RMS tumors, compared to FN RMS tumors [134]. In a panel of melanoma cell lines, the most prevalent alterations were found in *CDKN2A*, including copy number loss, methylation, or mutation in this gene, which resulted in a decreased or absent CDKN2A protein and increased palbociclib sensitivity [135]. In a breast cancer cell line, E2F2 overexpression was sufficient to bypass sensitivity to palbociclib [136]. Therefore, if E2F levels are high, cells can evade cell cycle arrest that is mediated by CDK4/6 inhibition [136]. Moreover, another study found that glioblastoma (GBM) xenograft lines with *CDKN2A* expression and either RB mutation or *CDK4* amplification were resistant to palbociclib. They suggested that CDK4 amplification leads to palbociclib resistance due to persistent RB phosphorylation by excessive CDK4. In addition, they demonstrated palbociclib sensitivity in an in vivo orthotopic GBM survival model. Palbociclib sensitivity was significant in this in vivo model with *CDKN2A/B*-deletion and CDK4 wild-type [137]. As part of another study, cell response to palbociclib was assessed on single-cell clones with high and low CDK2 activity sorted from triple-negative breast cancer cells. It was shown that the subpopulation of cells with high CDK2 activity bypassed the CDK4/6-dependency for cell cycle progression by the temporal deregulation of Cyclin E1 expression. However, the subpopulation of cells with low CDK2 activity could not overcome the inhibition of CDK4/6; therefore, they entered a quiescent state [138].

The underlying mechanisms that are associated with an acquired resistance to CDK4/6 inhibitors are discussed in the next sub section.

##### Acquired Mechanisms of Resistance to CDK4/6 Inhibitors and Possible Combination Therapies with CDK4/6 Inhibitors

Acquired resistance is the main reason for the therapeutic failure of CDK4/6 inhibitors in RB-proficient tumors, where tumor cells can acquire the ability to survive in the absence of CDK4/6 function [139]. Elucidation of the potential mechanisms of acquired resistance to CDK4/6 inhibitors may help to identify effective combination therapies to improve therapeutic responses to CDK4/6 inhibitors [87]. In contrast to cytotoxic drugs, CDK4/6 inhibitors mostly induce a cytostatic response by blocking CDK4/6-dependent cell cycle progression from the G1 to the S phase of the cell cycle [140]. The combination of CDK4/6 inhibitors with additional targeted therapies in RB-proficient tumors may change the cytostatic responses of CDK4/6 inhibitors to senescence and apoptosis [87]. Targeting major mitogenic signaling pathways such as the phosphatidylinositol 3’-kinase (PI3K) pathway or the mitogen-activated protein kinase (MAPK) pathway, along with CDK4/6 inhibition, may be a more promising combination for improving therapeutic efficacy (Figure 1) [87]. Moreover, both the PI3K and MAPK pathways are typically activated in cancer even before treatment. Therefore, it is possible that PI3K or MEK inhibitors, in combination with CDK4/6 inhibitors, are required to push the cells from cytostasis to cell death. For example, in estrogen receptor (ER)-positive breast cancer cell lines, upregulation of the PI3K pathway was identified as a consequence of chronic exposure to the CDK4/6 inhibitor palbociclib, which in turn upregulated Cyclin D1. In these cell lines, activated Cyclin D1 could drive cell cycle progression independently of CDK4/6 through the activation of CDK2; therefore, these cells could evade the cytostatic effect of the CDK4/6 inhibitor through noncanonical Cyclin D1-CDK2–mediated S-phase entry. In this study, the dual inhibition of the CDK4/6 and PI3K pathway prevented resistance to palbociclib both in vitro and in vivo [127]. In addition, it was shown that ribociclib, combined with PDK1 inhibitor GSK2334470 or the PI3K inhibitor alpelisib, decreased ER^+^ breast cancer xenograft tumor growth more efficaciously than either drug alone. In this study, primary in vitro studies showed the upregulation of phosphorylated and total PDK1 protein levels in ribociclib-resistant ER^+^ breast cancer cells, generated by chronic drug exposure [141]. PDK1 functions downstream of PI3K and activates AKT, where PDK1 at the plasma membrane binds to phosphatidylinositol-3,4,5-trisphosphate (PIP3), a product of PI3K. PIP3 phosphorylates and activates AKT [142]. The results of another study revealed that in palbociclib-resistant ER^+^ breast cancer cells, the Cyclin D-CDK4/6 pathway could be reactivated, but it remained sensitive to mTOR inhibitor vistusertib, suggesting that the inhibition of the PI3K pathway through the inhibition of mTOR may be a potential option for patients who have relapsed on therapy with CDK4/6 inhibitors. In this study, the combination of palbociclib with vistusertib resulted in more potent and durable regressions in breast cancer cell lines and xenografts [143]. Moreover, WES data on the biopsies from a melanoma patient showed that the oncogenic mutation PI3KCA^E545K^ pre-existed in a rare tumor cell subpopulation prior to combination therapy of the CDK4/6 inhibitor ribociclib and the MEK inhibitor MEK162. In this patient, it was demonstrated that these rare tumor cell subpopulations were positively selected and expanded early after ribociclib + MEK162 combination therapy. PI3KCA^E545K^ was identified as the only oncogenic mutation that confers resistance via increased activation of S6K1, a key downstream mTOR effector in the PI3K pathway. Therefore, an mTOR inhibitor could help to alleviate this resistance [144], possibly developed by CDK4/6 inhibitor ribociclib. As mentioned earlier, the MAPK signaling pathway can also contribute to CDK4/6 inhibitor resistance. In the following studies, the integration of an RNA sequencing analysis and phosphoproteomics profiling revealed that the MAPK pathway was hyperactivated in palbociclib-resistant prostate adenocarcinoma cell models. These palbociclib-resistant models were sensitized to MEK inhibitors, showing dependency on the active MAPK signaling pathway for tumor development when CDK4/6 was inhibited [145]. A study on bladder cancer also found that 995 genes were identified as significantly enriched after treatment with palbociclib in a bladder cancer cell line, using a genome-scale CRISPR-Cas9 activation screen. Significantly enriched related single-guide RNAs (sgRNAs) were randomly cloned and validated on both molecular and functional levels for mediating resistance to palbociclib. *MAP3K20* gene, a member of the MAPK signaling pathway, was among the validated sgRNAs considered as resistance mediators to palbociclib. In addition, the dual inhibition of the CDK4/6 and MAPK pathway exhibited synergistic effects in vitro and in vivo [146]. In agreement with these observations, the dual inhibition of the CDK4/6 and MAPK pathway has revealed beneficial preclinical outcomes in melanoma [147] and colorectal cancer [148,149,150]. These findings suggest that the dual inhibition of the CDK4/6 and PI3K or MAPK pathways may be more efficacious at overcoming resistance to CDK4/6 inhibitors and increasing their antitumor activity in patients who are sensitive to these inhibitors in the clinic.

Our data also support the efficacy of the dual inhibition of the CDK4/6 and PI3K or MAPK pathways in models of aggressive OS (Barghi et al., unpublished observations). Based on our in vitro screening data, the combination therapy of CDK4/6 and PI3K/mTOR inhibition induced a synergistic cell-growth inhibition that correlated with increased cell death in a panel of human RB+ OS cell lines. Moreover, our in vivo data demonstrated that in a pediatric patient-derived xenograft (PDX) model of relapsed OS that exhibited CDK4/6 hyperactivation, the dual inhibition of the CDK4/6 and PI3K/mTOR pathways was well tolerated and tumor growth was significantly decreased, compared to single agents (Barghi et al., unpublished observations).

## 5. Conclusions and Future Directions

Precision medicine in pediatric and AYA cancers, including sarcomas, seeks to identify therapeutic biomarkers that can reliably provide rationale and prioritization of the best therapy for each individual patient. To this end, the time of the biopsy collection and analysis may be critical. One can envision that the identification of high-risk signatures prior to front-line therapy is a strategy to pursue. Additionally, at relapsed stages, it is important to note that precision-guided treatment could be different from that at early stages. Therefore, obtaining another biopsy is appropriate for verifying previous actionable targets and/or identifying additional therapeutic biomarkers at the relapse stage [151]. To improve outcomes in pediatric and AYA sarcoma patients harboring poor prognostic signatures, it is essential to continue to develop and refine in-vitro and in-vivo modeling approaches that recapitulate not only the molecular signature of high-risk sarcoma patients, but also the tumor microenvironments, which can vary greatly in patients with aggressive sarcomas. To ultimately improve patient outcomes in the clinic, linking a patient’s clinical history to preclinical safety and efficacy data obtained from their PDX model is critical for studying pathogenesis and validating alternative or combination therapies. The next goal of precision medicine in pediatric and AYA cancer patients is to understand in detail the tumor adaptive response and develop combination therapies that can mitigate inhibitor resistance, such as that discussed on CDK4/6 inhibition. The development of resistance to CDK4/6 inhibitors from both the pre-activation of the PI3K or MAPK pathway or compensatory activation is being explored extensively in breast cancer, and clearly warrants preclinical testing in pediatric and AYA sarcomas with CDK4/6 hyperactivation. Importantly, continued approaches to understand and minimize toxic side effects need to receive more attention, as future therapies will move towards combinations of small molecule inhibitors [152]. In this article, we reviewed multiple pediatric and AYA precision medicine trials to highlight the dysregulation of the CDK4/6 cell cycle regulatory pathway in the most common types of pediatric and AYA sarcomas, as well as to provide insight into the promise and challenges of targeting the CDK4/6 pathway. In conclusion, the identification of potential therapeutic biomarkers by precision medicine has great potential to accelerate and advance successful patient outcomes, as well as develop less toxic treatments for pediatric and AYA cancer sarcoma patients. Thus, the establishment of precision medicine programs in academic centers and related clinical trials on pediatric and AYA sarcomas is the key to identifying and refining targeted therapies for these rare diseases, which, once classified into each subtype, represent even smaller patient populations [153].

## Figures and Tables

**Figure 1 cancers-14-03611-f001:**
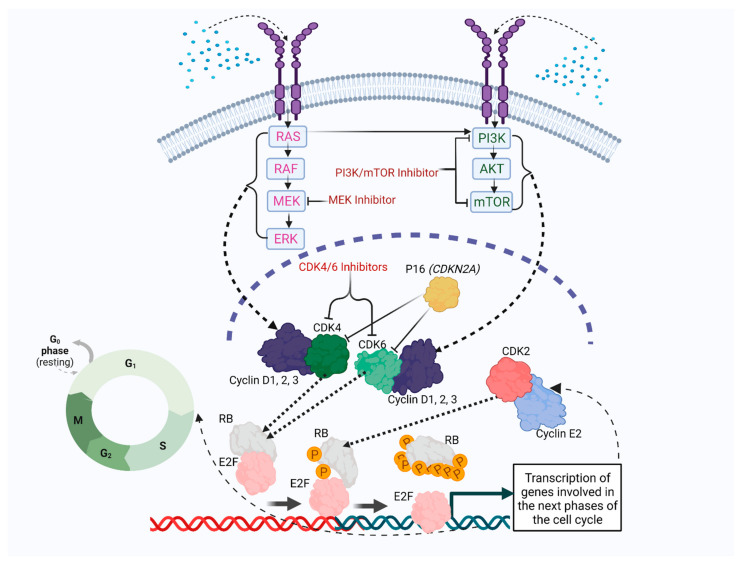
Targeting dysregulation of Cyclin D–CDK4/6 axis by CDK4/6 inhibition. Cyclin D-CDK4/6-mediated RB phosphorylation occurs upon activation of Cyclin D-CDK4/6 complex in response to extracellular signals such as stimulatory mitogens, leading to ensuing cell cycle progression. Initially, upon Cyclin D-CDK4/6-mediated RB phosphorylation, RB affinity to E2F is reduced, promoting transcription of genes such as Cyclin E2. Subsequently, the Cyclin E2-CDK2 complex hyperphosphorylates RB, which induces full dissociation of the RB-E2F complex and transcription of genes encoding proteins playing critical roles in the next phases of the cell cycle. The p16^INK4A^ (p16) tumor suppressor encoded by *CDKN2A* directly binds to CDK4/6 and inhibits the formation of the Cyclin D-CDK4/6 complex and prevents Cyclin D-CDK4/6-mediated RB phosphorylation. A dysregulated Cyclin D–CDK4/6 axis, which leads to uncontrolled cell proliferation, can be inhibited by CDK4/6 inhibitors. Chronic inhibition of CDK4/6 can trigger the upregulation of PI3K and MAPK pathways to compensate for inhibitory effects of CDK4/6 inhibitors on cell cycle progression by activation of D type cyclins. Dual inhibition of CDK4/6 and PI3K/mTOR or MEK may be an efficient combination treatment to prevent the compensatory effect of the PI3K or MAPK pathways on the development of CDK4/6 inhibitor resistance.

**Table 1 cancers-14-03611-t001:** Dysregulation of the CDK4/6 cell cycle regulatory pathway in pediatric and AYA OS, RMS and EWS patients.

Precision Medicine Trials	Genomic and Protein/RNA Biomarkers Associated with CDK4/6 Pathway		
	OS	RMS	EWS
INFORM [71]	Point mutations in *CDKN2B* and *CDK4* amplification	*CDKN2A/B* deletion or gain of *CDK4* copy number	*CCND2* amplification, *CDKN2A/B* deletion, *CCND1* overexpression
The Precision in Pediatric Sequencing (PIPseq) [72]	*RB* splice mutation *CCNE1* over-expression		
The individualized cancer therapy (iCat) [73]	*CCND1* and *CCNE1* high copy number gain	*CDKN2A/B* deletion	*CDKN2A* homozygous loss or single copy number gain of *CCND1*
The molecular biology tumor board (MBB) [75]	*CDKN2A* homozygous deletion	*CDK4* amplification	
TRICEPS [76]	*CDKN2A* copy loss/*CDKN2A* homozygous deletion and *RB* copy loss	*CDKN2A* point mutation, *RB* point mutation and loss of heterozygosity	
Phase 1 clinical trial program at MD Anderson Cancer Center [77]	*CDK4* amplification	*CDK4* amplification	*CDK4* amplification
MSKCC [78]	*CDK4* amplification and *CDKN2A* deletion/mutation		
The Zero Childhood Cancer Program [67]	CNV/SV of *CCNE1* and *RB* and *CCNE1* over-expression	CNVs of *CDK4, CDKN2A/B, CCND2;* or *CCND3* overexpression of *CDK4, CCND3,* or *CCNE1;* and down-regulation of *CDKN2A/B*	Homozygous deletion of *CDKN2A/B*
The St. Jude Children’s Research Hospital—Pediatric Cancer Genome Project [79]	SV of *CCND3* and *RB* point mutation/SV		
CRB cancer des Hôpitaux de Toulouse; BB-0033-00014 [80]	Stop-gained mutation in *RB*		
Complementary genomic study of OS patients [81]	*CDKN2A/B* deletions		
MOSCATO-01 [82]	*RB* deletion	*CDKN2A* deletion	*CDKN2A/B* deletions
Clin Omics Program [83]		*CDKN2A* and *RB* loss of heterozygosity	*CDKN2A* homozygous loss
RMS Case study [84]		*CDK4* amplification	
Molecular profiling of FET-TFCP2 RMS patients [57]		*CDKN2A* homozygous deletion	
Molecular profiling of EWS patients [85]			*CDKN2A* deletion

**Table 2 cancers-14-03611-t002:** Ongoing clinical trials on CDK4/6 inhibitors in pediatric and AYA sarcoma patients.

Disease	Treatment	Phase	ClinicalTrial.gov Identifier
Recurrent or refractory RB-positive solid tumors, including OS, EWS, and RMS	CDK4/6 inhibitor (palbociclib)	2	NCT03526250
Recurrent or refractory solid tumors, including EWS and RMS	CDK4/6 inhibitor (palbociclib) in combination with temozolomide and irinotecan, and/or with topotecan and cyclophosphamide.	2	NCT03709680
Recurrent or refractory RB-positive solid tumors, including OS, EWS, and RMS	CDK4/6 inhibitor (palbociclib) and other targeted therapies	2	NCT03155620
Soft tissue sarcomas, RMS	CDK4/6 inhibitor (ribociclib) in combination with doxorubicin hydrochloride)	1	NCT03009201
Recurrent or refractory solid tumors, including OS, EWS, and RMS	CDK4/6 inhibitor (abemaciclib)	1	NCT02644460
Soft tissue and bone sarcoma, including OS	CDK4/6 inhibitor (abemaciclib)	2	NCT04040205
EWS	CDK4/6 inhibitor (palbociclib) in combination with ganitumab	2	NCT04129151
Relapsed or refractory solid tumors, including OS, EWS, and RMS	CDK4/6 inhibitor (abemaciclib) in combination with irinotecan and temozolomide	1	NCT04238819

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
