# Peer review of "Precision Medicine Highlights Dysregulation of the CDK4/6 Cell Cycle Regulatory Pathway in Pediatric, Adolescents and Young Adult Sarcomas"

_cancers, 2022, doi:10.3390/cancers14153611_

Round 1
Reviewer 1 Report
In this review the authors provide discussed the clinical features and current therapies against sarcoma of children, adolescents, and young adult. They also have discussed roles of CDK4/6 in cell cycle regulation and relevance to cancer. They highlighted mechanisms of resistance to clinically approved CDK4/6 inhibitors and suggested potential combinations to overcome this issue. I believe the authors have addressed an important clinical challenge, sarcoma, and its management. The review reads well. I have general and specific comments and questions.
General comments
1. Across many pages, I see some lines and paragraphs have different font style and are written in bold letters. I don’t understand the relevance of this style of writing unless it is a typo error. I would suggest consistency in writing style.
2. When using the abbreviated form of CDK4 and CDK6 the authors used different forms, examples. CDK4/6: line 294, 295, 297, 362, 389, 452, 456, 470 and many other lines); CDK4/CDK6: lines 18, 29, 77, 309, 312, 383 and many lines; CDK4,6: lines 514, 516 and few other lines. This is unacceptable. You must be consistent in your writing of abbreviations, and I would suggest sticking to CDK4/6 than the others.
3. Line 29-30: the statement needs to be rewritten (it used double subjects)
4. Whenever writing genes, they should be in italics. Line 349 RB should be RB.
5. Line 691, CK4/6 shall be CDK4/6.
6. Figure 1: “transcription of genes involved in G1to S transition”. Better to change this to “transcription of genes involved in next phases of the cell cycle”
7. I feel statements written from line 618-624 are not related to the concepts discussed above, and I think it is better to incorporate these statements somewhere in 4.2.
Specific comments and questions
1. Line 504-505: “cell cycle is positively regulated by other cyclin-CDKs rather than cyclin D-CDK4/6”. What does this sentence mean? When saying rather than, is it referring negative role of CDK4/6 on cell cycle?
2. Is binding of D-type cyclins to CDK4/6 enough to fully activate the complex?
3. I believe you need to rewrite the part of your review which takes about role of cyclin D-CDK4/6 in the regulation of the cell cycle. Phosphorylation of serine and threonine residues of RB by activated cyclin D-CDK4/6 complex will not totally release E2F from the RB. Rather, phosphorylation of RB reduces its affinity for E2F and HDAC providing slight relief, and this enables the transcription of cyclin E2, which binds to CDK2. CDK2 by binding with cyclin E2 will further phosphorylate RB and enable complete release of E2F from RB, leading to transcription of genes coding for proteins such as cyclins A2, B1, TS, TK1, MCM2, PARP, DNMT1 etc which are important in next phases of the cell cycle. Moreover, cyclin D-CDK4/6 complex can bind to p21 (inhibitor of CDK1 and CDK2), and relieve CDK2, further enhancing its binding with cyclin E2.
4. Line 522-523: “CDK4/6 are promising targets for cancer treatment”. They are already well validated targets with the approval of clinically relevant inhibitors.
5. CDK4/6 can regulate cell cycle in ways not dependent on RB. For example, they can regulate FOXM1, SMAD2/3 transcription factors and regulate cell cycle. Hence, it is rational to include few points about RB-independent regulation of cell cycle by CDK4/6.
6. CDK4/6, though mostly play redundant roles, they also have distinct biological functions. Better to mention few points about some differential functions of CDK4/6.
7. When discussing 1st and 2nd generation inhibitors of CDKs, better to include examples from each class.
8. Authors claimed that to date there are 3 third generation CDK4/6 inhibitors. This is wrong. Currently, there are 4 inhibitors of CDK4/6 approved by US FDA: palbociclib, ribociclib, abemaciclib and trilaciclib. The first 3 are approved for the treatment of HR+ and HER2- advanced or metastatic breast cancer with anti-estrogens while the fourth one is used to prevent chemotherapy induced myelosuppression in the treatment of small cell lung cancer. A latest CDK4/6 inhibitor, dalpiciclib, has been approved by Chinese FDA for the treatment of HR+ and HER2- advanced or metastatic breast cancer with anti-estrogens. Therefore, I would strongly recommend the authors to correct accordingly.
9. It is better to mention some points on type and differences of toxicity profiles among approved inhibitors of CDK4/6.
10. Line 633, CDK4/6 inhibitors induce not only cytostatic response, but they could also induce permanent senescence and apoptosis depending on the type of cancer cell line used. Hence, better to state that “CDK4/6 inhibitors induce mostly cytostatic response”
Author Response
Reviewer 1
In this review the authors provide discussed the clinical features and current therapies against sarcoma of children, adolescents, and young adult. They also have discussed roles of CDK4/6 in cell cycle regulation and relevance to cancer. They highlighted mechanisms of resistance to clinically approved CDK4/6 inhibitors and suggested potential combinations to overcome this issue. I believe the authors have addressed an important clinical challenge, sarcoma, and its management. The review reads well. I have general and specific comments and questions.
General comments
- Across many pages, I see some lines and paragraphs have different font style and are written in bold letters. I don’t understand the relevance of this style of writing unless it is a typo error. I would suggest consistency in writing style.
We have made the font style consistent throughout each section.
- When using the abbreviated form of CDK4 and CDK6 the authors used different forms, examples. CDK4/6: line 294, 295, 297, 362, 389, 452, 456, 470 and many other lines); CDK4/CDK6: lines 18, 29, 77, 309, 312, 383 and many lines; CDK4,6: lines 514, 516 and few other lines. This is unacceptable. You must be consistent in your writing of abbreviations, and I would suggest sticking to CDK4/6 than the others.
We chose CDK4/6 throughout the review, as you suggested.
- Line 29-30: the statement needs to be rewritten (it used double subjects)
Line 29-30: the statement now has single subject.
- Whenever writing genes, they should be in italics. Line 349 RB should be
Now Line 282: RB is in italics.
- Line 691, CK4/6 shall be CDK4/6.
Now line 702: CK4/6 changes to CDK4/6
- Figure 1: “transcription of genes involved in G1to S transition”. Better to change this to “transcription of genes involved in next phases of the cell cycle”
In figure 1: transcription of genes involved in G1 to S transition” changed this to “transcription of genes involved in next phases of the cell cycle”
- I feel statements written from line 618-624 are not related to the concepts discussed above, and I think it is better to incorporate these statements somewhere in 4.2.
These statements are now moved up to line 511-518 in section 4.2. Thank you for your great suggestion!
Specific comments and questions
- Line 504-505: “cell cycle is positively regulated by other cyclin-CDKs rather than cyclin D-CDK4/6”. What does this sentence mean? When saying rather than, is it referring negative role of CDK4/6 on cell cycle?
Thank you for catching the confusing statement. We meant not only CDK4/6 but also other CDKs are positively regulate the cell cycle. Please refer to line 451-453 which the statement is revised.
- Is binding of D-type cyclins to CDK4/6 enough to fully activate the complex?
No, it is not enough! Please refer to line: 421-424
“The cyclins form a holoenzyme complex assembled with CDK4 or CDK6 (cyclin D‐CDK4/6 complex). The cyclin D‐CDK4/6 complex is not yet enzymatically fully activated unless by CDK activating kinase (CAK)-mediated activating phosphorylation of CDK4/6”
- I believe you need to rewrite the part of your review which takes about role of cyclin D-CDK4/6 in the regulation of the cell cycle. Phosphorylation of serine and threonine residues of RB by activated cyclin D-CDK4/6 complex will not totally release E2F from the RB. Rather, phosphorylation of RB reduces its affinity for E2F and HDAC providing slight relief, and this enables the transcription of cyclin E2, which binds to CDK2. CDK2 by binding with cyclin E2 will further phosphorylate RB and enable complete release of E2F from RB, leading to transcription of genes coding for proteins such as cyclins A2, B1, TS, TK1, MCM2, PARP, DNMT1 etc which are important in next phases of the cell cycle. Moreover, cyclin D-CDK4/6 complex can bind to p21 (inhibitor of CDK1 and CDK2), and relieve CDK2, further enhancing its binding with cyclin E2.
We appreciate your feedback on this very important section of the review. We revised this part of the review based on your valuable comments. Accordingly, we have revised the figure 1 as well. Please refer to line: 443-453
“E2F transcriptional activity is not only inhibited by the direct binding to RB but also by RB-mediated recruitment of histone deacetylase (HDAC) to the RB-E2F complex (Taylor et al). Initially, upon cyclin D-CDK4/6-mediated RB phosphorylation, RB affinity to E2F and HDAC is reduced, promoting transcription of genes such as Cyclin E2 (Taylor et al). Subsequently, the Cyclin E2-CDK2 complex hyperphosphorylates RB which induces full dissociation of the RB-E2F complex and transcription of genes encoding proteins playing critical roles in the next phases of the cell cycle, such as cyclin A2, cyclin B1 and others involved in DNA replication and repair. (Taylor et al, Zhu et al and ref 94 (Braden et al)). Cyclin E2-CDK2 exemplifies that the cell cycle is not only positively regulated by cyclin D-CDK4/6 but also by other cyclin-CDKs (Figure 1).”
New references used:
The Role of Rac and Rho in Cell Cycle Progression - ScienceDirect
E2Fs link the control of G1/S and G2/M transcription - PMC (nih.gov)
- Line 522-523: “CDK4/6 are promising targets for cancer treatment”. They are already well validated targets with the approval of clinically relevant inhibitors.
Now line 480: promising changed to well validated.
- CDK4/6 can regulate cell cycle in ways not dependent on RB. For example, they can regulate FOXM1, SMAD2/3 transcription factors and regulate cell cycle. Hence, it is rational to include few points about RB-independent regulation of cell cycle by CDK4/6.
Statements on FOXM1 and SMAD2/3 have been included in lines: 455-462.
“CDK4/6 can also regulate the cell cycle independently of RB. CDK4/6 initiates phosphorylation of FOXM1 in G1, leading to transcriptional activation of FOXM1, which is required for the expression of genes involved in cell cycle progression (Andres et al). CDK4 negatively regulates SMAD2/3, components of the TGFβ signaling cascade that exhibit anti-proliferative activity (Matsuura et al). Moreover, CDK4-mediated phosphorylation of SMAD3 leads to its inhibition and release of G1 arrest. Therefore, diminishing SMAD3 activity by CDK4-mediated phosphorylation could contribute to tumorigenesis (Zelivianski et al).”
New references used:
A systematic screen for CDK4/6 substrates links FOXM1 phosphorylation to senescence suppression in cancer cells - PubMed (nih.gov)
Cyclin-dependent kinases regulate the antiproliferative function of Smads - PubMed (nih.gov)
CDK4-Mediated Phosphorylation Inhibits Smad3 Activity in Cyclin D Overexpressing Breast Cancer Cells - PMC (nih.gov)
- CDK4/6, though mostly play redundant roles, they also have distinct biological functions. Better to mention few points about some differential functions of CDK4/6.
We have included some details on differential functions of CDK4 and CDK6. Please refer to line: 426-433.
“However, it has been shown that CDK4 and CDK6 can differentially regulate cancer progression through various mechanisms in breast, prostate, and pancreatic cancer models- CDK4-dependent regulation of prometastatic inflammatory pathways, and CDK6-dependent control of genes involved in DNA replication and repair mechanisms. In these models, silencing CDK6 but not CDK4 led to induction of DNA damage and defective DNA repair. In contrast, silencing CDK4 but not CDK6 enriched for the TNF signaling pathway involved in regulation of proinflammatory cytokines and chemokines”
- When discussing 1stand 2nd generation inhibitors of CDKs, better to include examples from each class.
Please see Line 488-492 where we have included examples from each class.
“The firstgeneration pan-CDK inhibitors such as flavopiridol and roscovitine lacked selectivity among the CDK family members (101). The second-generation including, dinaciclib, P276-00, AT7519, TG02, roniciclib, RGB-286638 were developed based on the first-generation group but with increasing selectivity against CDK1 and CDK2 to reduce off-target risks of CDK inhibitors.”
- Authors claimed that to date there are 3 third generation CDK4/6 inhibitors. This is wrong. Currently, there are 4 inhibitors of CDK4/6 approved by US FDA: palbociclib, ribociclib, abemaciclib and trilaciclib. The first 3 are approved for the treatment of HR+ and HER2- advanced or metastatic breast cancer with anti-estrogens while the fourth one is used to prevent chemotherapy induced myelosuppression in the treatment of small cell lung cancer. A latest CDK4/6 inhibitor, dalpiciclib, has been approved by Chinese FDA for the treatment of HR+ and HER2- advanced or metastatic breast cancer with anti-estrogens. Therefore, I would strongly recommend the authors to correct accordingly.
We appreciate you making us aware of the other CDK4/6 inhibitors, trilaciclib and dalpicclib. In the mentioned lines below, you can find info on the two additional CDK4/6 inhibitors.
Line: 510-511
“Trilacicilb is the most recently CDK4/6 inhibitor approved by U.S. FDA in 2021 (Powell).”
Line: 523-524
“Biochemical profiling of trilaciclib demonstrated similar potency toward CDK4/cyclin D1 and CDK6/cyclin D3 (Bisi et al).”
Line: 549-559
“Trilaciclib is well tolerated in the patients, and its administration before chemotherapy is not associated with a clinically relevant toxicity increase and significantly reduces chemotherapy-related toxicity (Weiss et al). To date, there are no clinical trials to investigate the efficacy and safety of trilaciclib in pediatric and AYA sarcomas. In addition to four U.S. FDA approved CDK4/6 inhibitors, there is a novel CDK4/6 inhibitor, dalpiciclib (SHR6390), approved by China’s National Medical Products Administration (NMPA) for the treatment of HR+ and HER2- recurrent or metastatic breast cancer with endocrine agents (Fan et al & Zhang et al). Dalpiciclib exhibits a similar potency against CDK4 and CDK6 (Zhang et al). The oral administration of dalpiciclib with dosing every three weeks with a one-week break in patients with advanced breast cancer was well-tolerated (Zhang et al).”
New references used:
Concerning FDA approval of trilaciclib (Cosela) in extensive-stage small-cell lung cancer - PubMed (nih.gov)
Preclinical Characterization of G1T28: A Novel CDK4/6 Inhibitor for Reduction of Chemotherapy-Induced Myelosuppression - PubMed (nih.gov)
Effects of Trilaciclib on Chemotherapy-Induced Myelosuppression and Patient-Reported Outcomes in Patients with Extensive-Stage Small Cell Lung Cancer: Pooled Results from Three Phase II Randomized, Double-Blind, Placebo-Controlled Studies - PubMed (nih.gov)
A phase 1 study of dalpiciclib, a cyclin-dependent kinase 4/6 inhibitor in Chinese patients with advanced breast cancer - PubMed (nih.gov)
Current clinical trials on breast cancer in China: A systematic literature review - PubMed (nih.gov)
- It is better to mention some points on type and differences of toxicity profiles among approved inhibitors of CDK4/6.
This is very good point. Thank you for this great suggestion!
Please find the info lines: 533-545
“Despite the similar mechanism of action for the first three CDK4/6 inhibitors, palbociclib, abemaciclib, and ribociclib, they exhibit different toxicity profiles and dose delivery schedules. Since half-lives of palbociclib and ribociclib are more than 24 hours, they are dosed daily. Due to their myelosuppressive effects, daily dosing is non-continuous, 21 days followed by a break of one week for neutrophil count recovery in treated patients. In comparison, the half-life of abemaciclib is <24 hours, induces relatively less myelosuppression, and is dosed twice daily (George et al). Palbociclib and ribociclib have minimal GI toxicities compared to abemaciclib, with extensive GI toxicities (Rugo et al). However, abemaciclib-mediated bone marrow toxicity is less serious than palbociclib and ribociclib-mediated bone marrow toxicities (George et al). The rate of hematological toxicity, mainly neutropenia, is higher for palbociclib and ribociclib (Costa et al). Moreover, it has been shown that ribociclib can induce hepatic toxicity (Onesti et al).”
New references used:
Clinical and Pharmacologic Differences of CDK4/6 Inhibitors in Breast Cancer - PubMed (nih.gov)
Management of Abemaciclib-Associated Adverse Events in Patients with Hormone Receptor-Positive, Human Epidermal Growth Factor Receptor 2-Negative Advanced Breast Cancer: Safety Analysis of MONARCH 2 and MONARCH 3 - PubMed (nih.gov)
Meta-analysis of selected toxicity endpoints of CDK4/6 inhibitors: Palbociclib and ribociclib - PubMed (nih.gov)
CDK4/6 inhibitors in breast cancer: differences in toxicity profiles and impact on agent choice. A systematic review and meta-analysis - PubMed (nih.gov)
- Line 633, CDK4/6 inhibitors induce not only cytostatic response, but they could also induce permanent senescence and apoptosis depending on the type of cancer cell line used. Hence, better to state that “CDK4/6 inhibitors induce mostly cytostatic response”
Please see line: 642-644 where we have revised the statement based on your comment.
“In contrast to cytotoxic drugs, CDK4/6 inhibitors induce mostly cytostatic response by blocking CDK4/6-dependent cell cycle progression from G1 to S phase of cell cycle”
Submission Date
06 June 2022
Date of this review
17 Jun 2022 04:56:18

Reviewer 2 Report
This is a very nice paper. Congratulations.
One major comment: as Cancers is an oncology targeted paper, I find the cca. 240-line Introduction quite lengthy (more than 1/3) , which is not directly connected to the main issue of the paper. I suggest the significant shortening of the lengthy introduction (maximum 1/3 of the original would be left).
Minor: several mistypes
Author Response
Reviewer 2
This is a very nice paper. Congratulations.
One major comment: as Cancers is an oncology targeted paper, I find the cca. 240-line Introduction quite lengthy (more than 1/3), which is not directly connected to the main issue of the paper. I suggest the significant shortening of the lengthy introduction (maximum 1/3 of the original would be left).
Minor: several mistypes
Thank you for your helpful suggestion. We have substantially reduced this section by over 50% by removing epidemiology, clinical features and related predisposition syndromes. As background for the reader, we now mention these references in lines: 93-94. We streamlined the formatting throughout the manuscript.
Submission Date
06 June 2022
Date of this review
23 Jun 2022 17:19:45

Reviewer 3 Report
The authors selected an interesting topic and their review was very valuable. However, the decision to include a general review of sarcomas then proceed to describe dysregulation of the CDK4/CDK6 pathway, lengthened the article unnecessarily. There was a fair amount of redundancy that could have been avoided by summarizing the first five pages of the article in a table. There were multiple formatting errors showing different texts used in writing the manuscript. The table formatting would benefit from attempting to include the table in one rather than 2 pages.
Author Response
Reviewer 3
The authors selected an interesting topic, and their review was very valuable. However, the decision to include a general review of sarcomas then proceed to describe dysregulation of the CDK4/CDK6 pathway, lengthened the article unnecessarily. There was a fair amount of redundancy that could have been avoided by summarizing the first five pages of the article in a table. There were multiple formatting errors showing different texts used in writing the manuscript. The table formatting would benefit from attempting to include the table in one rather than 2 pages.
Thank you for your input. We have substantially reduced this section by removing epidemiology, clinical features and related predisposition syndromes. As background for the reader, now mention these references in lines: 91-94. We streamlined the formatting throughout the manuscript.

Round 2
Reviewer 1 Report
The authors addressed all my comments and suggestions vey well. I particularly like the way you modified figure 1.
Reviewer 3 Report
The authors have mad manuscript modifications that addressed my comments on former review. I have no objection to publication.